# Quantification of Thoracic Volume and Spinal Length of Pediatric Scoliosis Patients on Chest MRI Using a 3D U-Net Segmentation

**DOI:** 10.3390/healthcare13182327

**Published:** 2025-09-17

**Authors:** Romy E. Buijs, Dingina M. Cornelissen, Dimo Devetzis, Peter P. G. Lafranca, Daniel Le, Jiaxin Zhang, Mitko Veta, Koen L. Vincken, Tom P. C. Schlösser

**Affiliations:** 1Medical Imaging Master’s Programme, Utrecht University, 3584 CS Utrecht, The Netherlands; r.e.buijs@students.uu.nl (R.E.B.); d.m.cornelissen@students.uu.nl (D.M.C.); d.devetzis@student.tue.nl (D.D.); j.zhang17@students.uu.nl (J.Z.); 2Medical Imaging Master’s Programme, Eindhoven University, 5612 AZ Eindhoven, The Netherlands; 3Department of Orthopedic Surgery, UMC Utrecht, P.O. Box 85500, 3508 GA Utrecht, The Netherlands; p.p.g.lafranca-2@umcutrecht.nl; 4Medical Image Analysis Group, Eindhoven University of Technology, 5612 AZ Eindhoven, The Netherlands; m.veta@tue.nl; 5Department of Radiology, Image Sciences Institute, UMC Utrecht, 3508 GA Utrecht, The Netherlands; k.vincken@umcutrecht.nl

**Keywords:** adolescent idiopathic scoliosis, chest MRI, chest volume, spinal length, chest deformation, automatic segmentation, 3D U-Net

## Abstract

**Background/Objectives:** Adolescent idiopathic scoliosis (AIS) can lead to significant chest deformations. The quantification of chest deformity and spinal length could provide additional insights for monitoring during follow-up and treatment. This study proposes a 3D U-Net convolutional neural network (CNN) for automatic thoracic and spinal segmentations of chest MRI scans. **Methods:** In this proof-of-concept study, axial chest MRI scans from 19 girls aged 8–10 years at risk for AIS development and 19 asymptomatic young adults were acquired (n = 38). The thoracic volume and spine were manually segmented as the ground truth (GT). A 3D U-Net CNN was trained on 31 MRI scans. The seven remaining MRI scans were used for validation, reported by the Dice similarity coefficient (DSC), the Hausdorff distance (HD), precision, and recall. From these segmentations, the thoracic volume and 3D spinal length were calculated. **Results:** Automatic chest segmentation was possible for all chest MRIs. For the chest volume segmentations, the average DSC was 0.91, HD was 51.89, precision was 0.90, and recall 0.99. For the spinal segmentation, the average DSC was 0.85, HD was 25.98, precision was 0.74, and recall 0.99. Chest volumes and 3D spinal lengths differed by on average 11% and 12% between automatic and GT, respectively. Qualitative analysis showed agreement between the automatic and manual segmentations in most cases. **Conclusions:** The proposed 3D U-Net CNN shows a high accuracy and good predictions in terms of HD, DSC, precision, and recall. This suggested 3D U-Net CNN could potentially be used to monitor the progression of chest deformation in scoliosis patients in a radiation-free manner. Improvement can be made by training the 3D U-net with more data and improving the GT data.

## 1. Introduction

Adolescent idiopathic scoliosis (AIS) is a three-dimensional (3D) deformity of the spine and trunk which occurs in 2–3% of the healthy population during the adolescent growth spurt [1,2,3]. Progressive scoliosis can lead to significant chest deformation, with the narrowing of the ipsilateral hemithorax, a rib hump, and spinal intrusion into the chest. This eventually can lead to restrictive and obstructive pulmonary impairment [4,5]. The progression of the spinal deformity during growth is generally monitored using full-spine radiographs [6]. The quantification of the chest deformity, chest volume, and 3D spinal length could provide additional insights to determine the severity of thoracic deformation and the effect of a brace or operative treatment on the chest [7,8]. Currently, there is no routine 3D imaging of the chest deformity in scoliosis patients, as no efficient acquisition and measurement method is available. For research purposes, chest deformation has been quantified on reconstructions of biplanar radiographs, computed tomography (CT) scans, and magnetic resonance imaging (MRI). Biplanar radiograph reconstructions are time-consuming and CT scans expose the patient to ionizing radiation which, especially in young patients, increases the risk of developing cancer [9,10,11,12,13,14,15,16,17]. MRI is acquired in a radiation-free way and the chest deformity parameters can be extracted from segmentations of the chest wall and spine created from the 3D images [18]. However, to speed up this process, the segmentation process ideally needs to be automatic. There is a knowledge gap in the best way to use automatic MRI segmentations for assessing chest deformity.

Recently, new techniques for automatic and accurate segmentations have become more commonly used in 3D medical image processing. The 3D U-Net convolutional neural network (CNN) is a commonly used architecture that has three encoding and decoding blocks in the down-sampling and up-sampling path, respectively [19,20,21]. The encoding or down-sampling path captures image features through convolutional and max-pooling layers, while the decoding or up-sampling path reconstructs from the compressed representation using transpose convolution layers combined with skip connections. Skip connections preserve spatial information by concatenating low-level feature maps with high-level feature maps. The model learns the context of relative information between voxels in order to output the desired segmentation mask. While the segmentation task can potentially be performed in a 2D slice-by-slice manner, it can be argued that a 3D model may perform better due to its ability to capture information in three dimensions. To the best of our knowledge, there is no 3D U-Net CNN that uses MRI images to create automatic segmentations for quantifying thoracic volume and spinal length.

The aim of this study was to test if axial chest MRIs can be used to build a 3D U-Net, in order to automatically quantify thoracic volume and spinal length during growth. We present a proof of concept of a fully automated image processing technique that could facilitate the quantification of these parameters in future larger datasets.

## 2. Materials and Methods

### 2.1. Data Collection

The data included 19 T2-weighted (T2w) chest MRIs from the younger sisters of scoliosis patients between 8 and 10 years old and 19 MRIs from young adult volunteers. Subjects could be included if they had no medical history of spinal or neuromuscular disease. The younger sisters were part of the baseline of the prospective EARLYBIRD study; more detailed inclusion and exclusion criteria can be found in this study [22]. Data was obtained after the written informed consent of the participant and/or legal guardians. This study has been approved by the Medical Ethics Committee NedMed and registered at clinicalstrials.gov (NCT05924347). For the purpose of chest deformity quantification, T2w axial MRIs from T1 to T12 were obtained in addition to spinal MRIs (Philips 1.5 Tesla Achieve dStream, Philips Healthcare, Best, The Netherlands). The axial series included the screening of the whole chest, with a slice thickness of 4 mm and gaps ranging from 20 to 25 mm, with the total number of axial plane slices varying from 12 to 17 slices. Pixel spacing ranged from 0.46875 to 0.625 mm. All pixels were isotropic.

The thoracic volume was defined from the most cranial slice containing the sternum to that containing the top of both kidneys. In case of abnormal kidney position, it was discussed with an orthopedic surgeon which slice was used as the most caudal slice. The thoracic spine was defined from the most cranial slice containing the sternum to T12, identified by the presence of the median arcuate ligament, and its shape and relation to the ribs. Using these boundaries, five operators manually segmented the thoracic volume within the chest wall and the spine, including vertebral bodies, intervertebral disks, transverse, and spinous processes on all 38 scans using 3D Slicer (v5.4.0, Brigham and Women’s Hospital, Boston, MA, USA). One operator reviewed all spine segmentations for consistency.

From all 38 available scans, 31 were used to train the 3D U-Net, while the 7 other scans were used for the validation of the 3D U-Net. It was made sure that both the training and test data had a balanced ratio of children and young adults. Training data was augmented using a −10° to 10° rotation along the longitudinal axis on all slices of each scan. The study was intended as a proof of concept.

### 2.2. 3D U-Net

For the semantic segmentation task, a 3D U-Net model was applied using PyTorch (v2.0.1, Meta AI, Menlo Park, CA, USA). The model received a grayscale 3D image as input represented by a tensor of shape (1, 240, 240, 16), where the dimensions correspond to (Channel, Height, Width, Depth) with height, width, and depth expressed in number of pixels. The model was trained using the manually segmented scans as ground truth, learning to classify each voxel into one of three categories: background, thoracic volume, or spine. The output was a tensor of shape (3, 240, 240, 16), where each channel contained the probability of a voxel belonging to each class. Each voxel was assigned to the category with the highest probability.

To fit the dimensions of the input tensor, all MRI scans were min–max normalized between 0 and 1 and bicubic interpolation was performed. Similarly, the manual segmentation mask was made to fit using nearest-neighbor interpolation.

The acquired segmentation mask from the 3D U-Net is sampled back to its original number of slices using nearest-neighbor interpolation. A 3D median filter of size (5, 5, 5) was applied to smoothen the edges between the slices. The resulting 3D mesh of volume and the spine are visualized in Figure 1a.

### 2.3. Three-Dimensional Spinal Length Quantification

To quantify the 3D length of the thoracic spine within the thorax from the spine mask, the center of mass (COM) of the spine mask for each slice was used as the consistent landmark. The distances between the COMs of adjacent slices were calculated and corrected for physical spacings. These corrected distances were summed to obtain the 3D spinal length. The COM method was chosen because it is easily computable and offers a uniform definition for each vertebra given that each is symmetrical. This allows a basic comparison of spinal length between the model output and the manual segmentations.

### 2.4. Validation Metrics

For every slice, two heat maps were created, reflecting the probability of each voxel belonging to the thoracic volume and spine, as given by the output of the model. A value of 1 indicated complete certainty that the voxel belonged to the class, while a value of 0 indicated absolute exclusion.

The automatic segmentation model was evaluated and validated on the independent test set using Python (v3.11, Python Software Foundation, Wilmington, DE, USA). The aim of this validation was to score the similarity between the predicted segmentation and the annotated ground truth segmentation. The following quantitative performance metrics were used [23]:Dice similarity coefficient (DSC): two times the overlap between the predicted segmentation and the ground truth segmentation over the total area of the predicted and ground truth segmentation.Hausdorff’s distance (HD): the Euclidean distances between the prediction and ground truth boundaries. Additionally, the 95th percentile of the Hausdorff distance (95HD) is considered, which disregards large outliers.Precision or positive predictive value: defined as the ratio of true positives to the total number of positive predictions.Recall or true positive rate: the ratio of correctly predicted pixels corresponding to volume to the total number of ground truth volume pixels.

## 3. Results

### 3.1. Quantitative Analysis

The quantitative performance metrics of all test subjects that were included in the validation dataset are shown in Table 1 and Table 2. The predicted volumes for adult volunteers were bigger than children (EBS group). The segmentation of thoracic volume had a mean DSC of 0.91 (range: 0.87–0.94), mean precision of 0.90 (range: 0.78–0.96), and recall was 0.99 (range: 0.96–1.00). The average HD was 51.89 mm (range: 25.50–179.41 mm), with a 95HD of 12.47 mm (range: 1.67–27.19 mm).

The high recall indicates near-complete chest and spinal structure segmentations, but the lower DSC and precision suggest that some additional volumes were (false positively) included for both parameters. The HD and 95HD indicates this is mainly due to sharp outliers.

Table 3 and Table 4 show the model’s output of chest volume and spinal length compared with the ground truth. On average, the model overestimates the thoracic volume by 11%. For spinal length, an average overestimation of 12% was observed. In all but one case were chest volume and spinal length overestimated, with the largest outlier being 29%. For both thoracic volume and 3D spinal length, the model volume yielded higher estimates, with (Cohen’s *dz* = 1.01, 95% CI [0.08–1.95]) and (Cohen’s *dz* = 1.07, 95% CI [0.12–2.03]) for both, respectively. The small sample size was reflected by the wide confidence intervals.

### 3.2. Qualitative Analysis

Consistent with the results of the quantitative analysis, the model accurately predicts the segmentation of most slices. Figure 1 shows an example of a set of slices with thoracic volume and spine predictions. For most slices, the predictions of the 3D U-Net model agree very well with the manual segmentations, both for thoracic volume and 3D spinal length. Figure 2 shows an overview of three different subjects: the top row shows an accurate segmentation, with relatively low over-segmentation. The middle row shows a predicted volume with a relatively inaccurate shape, i.e., over-segmentation. The bottom row shows that all voxels were correctly predicted as background as the area lies outside the region of interest (ROI). The latter also highlights the model’s ability to correctly predict when a prediction mask is or is not required, demonstrating robustness.

## 4. Discussion

The aim of this study was to quantify thoracic volume and spinal length using a 3D U-net convolutional neural network. The results show that after training, the model can estimate the volume of the thorax and the length of the spine with reasonable to high accuracy in terms of the Hausdorff distance, Dice similarity coefficient, precision, and recall. In terms of the percentage difference in chest volume and spinal length from manual segmentations, the differences showed a wide range between −2.0% and +29.4%.

When comparing the results to prior studies that use deep learning for vertebrae segmentation, it appears that our model achieves a performance which comes very close to other, state-of-the art models. For instance, Lu et al. (2023) used a similar U-Net structure to segment lumbar vertebrae from CT images and only scored a little better, achieving Dice scores ranging from 0.89 to 0.91 when validated on a public dataset compared with our scores ranging from 0.79 to 0.88 [24]. In this study, the training dataset was much larger, with 400 available CT scans compared with our 31. The difference in the size of the training dataset could explain the higher Dice scores. In general, studies that use large datasets (>400 images) to train vertebrae segmentation models show higher Dice scores. For example, two other recent studies that used larger training datasets showed Dice scores of around 0.950 [25,26]. Our model shows, with a very small training dataset, already-satisfactory Dice scores, and therefore seems promising, with possible higher Dice scores after being trained on even larger datasets.

Notably, our MRI images contain wide inter-slice gaps (20–25 mm), so consecutive axial slices share little anatomical continuity, limiting the contextual information available to a 3D U-Net and therefore constraining the Dice scores. By contrast, Lu and Saeed processed regular diagnostic CTs, making neighboring slices easier to correlate and thus boosting segmentation performance [24,26]. Suri et al. (2021) used MRI images but adopted a purely 2D architecture, where the slice gap has no influence on its performance [25]. These differences also offer a plausible explanation for the modest Dice score gap between our results and those reported in the literature.

Whereas the majority of studies on vertebral segmentation focus on the automatic segmentation of CT images, our model distinguishes itself because it is suited for MR images. MRI is radiation-free, offering great benefits compared with CT or radiography, especially for young patients where radiation exposure has been shown to increase cancer risk [9,10,13,15,16,17]. In general, practitioners strive to keep radiation for patients as low as reasonably achievable (ALARA principle). The model created in this study offers a completely radiation-free alternative, with its accuracy being almost as good as CT-based models. Another advantage of our model is that it simultaneously segments the thorax and spine and calculates the volume and length, respectively. This functionality is especially useful in scoliotic patients, since it provides direct insight into the severity of the scoliotic curve and its effect on the thoracic volume. This feature makes the model easily applicable in practice, since the model output is intuitive and easy to obtain without profound knowledge of the model architecture. To our knowledge, there are currently no other existing models with the same functionality. Moreover, an MRI scan can be used for screening for neural abnormalities in AIS [27]. Recently, an MRI sequence has been developed which can be used to create MRI-based, AI-generated synthetic CT images [28,29,30]. These synthetic CT images can be used to assess scoliosis parameters [31,32]. It has also recently been shown that these synthetic CT images can be used for navigated pedicle screw placement, a procedure needed to correct severe scoliosis cases [33]. When this MRI sequence is added to a scan, both information on the thorax and spine and CT-like images can be obtained in just one MRI session.

Although the spinal length and thoracic volume were predicted with adequate accuracy for the majority of the participants, in a small number of cases the model mispredicts the volume or the spinal length. Firstly, there were some false positives on slices without a manual mask, with 13 out of all 112 slices (of the seven tested volumes) having this wrong prediction. This problem was most prominent in one EBS subject and one volunteer, which could partially explain the low volume segmentation precision values for these two subjects compared with those of the other subjects. Secondly, the segmentation of an EBS subject had a high Hausdorff’s distance, which indicates that the model made a prediction on a slice (slice 2) far away from our manual segmentation (slice 9 to 14). These first two types of outliers show that the model has difficulty in determining the boundaries of the thoracic cage and the spine, which could result from the relatively large gap between slices in the original data. The third type of outlier is an inaccurate shape prediction. Three slices show this type of outlier, all from the same volunteer.

These outliers indicate that the model overestimates the thoracic volume and spinal length compared with manual segmentations. This overestimation is particularly caused by difficulties in defining the lower bound of the segmentation. We think this could be improved by using a more pronounced marker for the lower bound. For the definition of the lower bound of the segmentations, vertebra T12 was chosen. Since pinpointing the exact location of the T12 on a T2w MRI proved to be challenging, the top of the kidneys was used for determining the lower limit of thoracic volume. However, the location of the kidneys is not the same slice as the location of T12 for every person and the kidneys do not keep the same position relative to the spine during growth. For spinal length, to determine the lower border we used the difference in vertebral shape of the T12 (thoracic vertebra) versus the shape of the L1 (lumbar vertebra). Additionally, we looked at the location of the median arcuate ligament, since it is located at the T12-L1 level. However, this ligament is not always clearly visible in T2w MRIs and moves caudally with inspiration [34]. These unclear definitions of the lower bound for both thoracic volume and spinal length possibly caused some error and thereby decreased the accuracy of the model. Therefore, a more solid definition could improve the results.

The automatic segmentation of the spine proved to be more challenging than the segmentation of the thorax. This could be caused by the fact that the vertebrae, especially its processes, are smaller and more refined than the thoracic cavity. Additionally, the exact outlines of the vertebrae are difficult to distinguish on a T2w MRI scan, both on automatic and manual segmentations. For spinal length calculation, the COM of each vertebra was used. This approach, however, depends on consistent slice selection. The MRI gaps of 20–25 mm result in selecting either intervertebral disks or vertebrae at different heights. Since intervertebral disks lack spinous processes, this significantly shifts the COM. Additionally, the COM method relies on the angle of the slice through the vertebrae. In scoliosis, concurrent kyphosis causes the vertebrae to appear longer at the kyphotic region. We suggest that in an updated model, instead of COM, the anterior and posterior vertebral points of the vertebra should be used instead of the COM.

Lastly, the performance of the model is limited by the amount of training data that is available. In this project there were 39 data points in total that were split into training, validation, and test sets. Next to this small sample size, there was also a mismatch between the two groups (children of 8–10 years and adult volunteers). To reach the optimal performance of the model, ideally the training set would consist of >1000 images, including those of some AIS patients. This would improve the model significantly.

Improving the model with the aforementioned adjustments could save time on the analysis of image results for radiologists. Although the results of the model should always be checked for accuracy, this model allows for the calculation of important parameters in scoliosis without the need for manual segmentation.

## 5. Conclusions

The proposed 3D U-Net CNN was developed for the automatic segmentation of the spine and chest on axial T2w chest MRI acquired for the purpose of chest wall deformation quantification in spinal deformity patients. Especially considering the limited amount of data available for training, the model shows reasonable to high accuracy. Good predictions were shown in terms of HD, DSC, precision, and recall. With further training and more precise attention to the labeling of the model’s input data, the performance of the model can be improved to make the model’s estimations more accurate. Since this analysis is performed using non-invasive T2w MR imaging, our suggested 3D U-Net CNN seems a promising proof of concept, but needs further validation before use in clinics to automatically quantify the thoracic volume and 3D spinal length in order to monitor the progression of 3D chest deformation in children and young adults with spinal deformities in a radiation-free manner.

The take-home messages are as follows:-A 3D U-net CNN using MRI images to quantify thoracic volume and spinal length could successfully be created.-Our CNN showed reasonably high accuracy, but an overestimation of volumes and length.-This proof of concept needs further training on bigger datasets before use in clinics.

## Figures and Tables

**Figure 1 healthcare-13-02327-f001:**
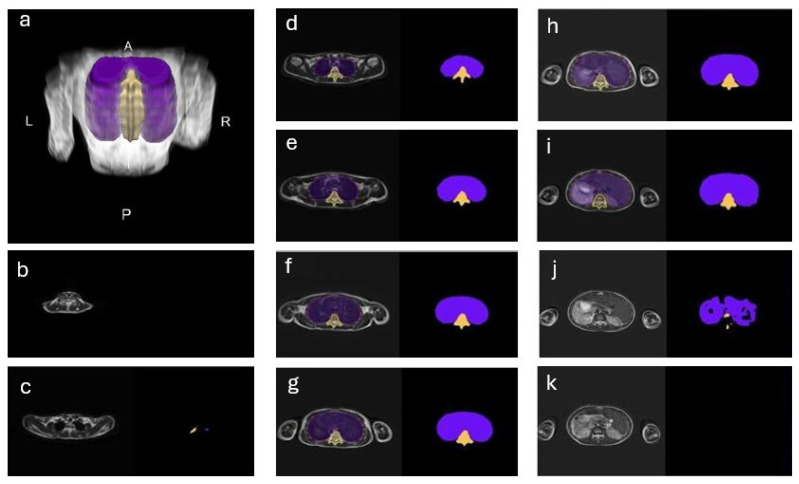
Segmentations of the thoracic volume (purple) and spine (yellow). (**a**) 3D mesh representation of thoracic volume and spine segmentation using the resampled predicted masks. A: anterior; P: posterior; R: right; L: left. (**b**–**k**) Axial MRI slices from cranial (**b**) to caudal (**k**), showing the original MRI with manual segmentation overlay (left) and the predicted mask (right).

**Figure 2 healthcare-13-02327-f002:**
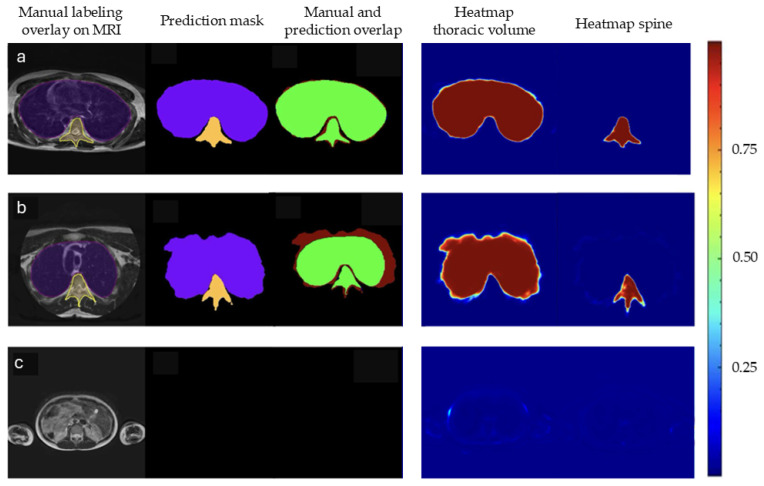
(**a**) Visual agreement between manual and predicted segmentations: participant 2, slice 12. (**b**) Over-segmentation: participant 1, slice 15. (**c**) No foreground structures on both manual and predicted segmentations: participant 5, slice 6. The first column shows manual segmentations of the thoracic volume (purple) and spine (yellow) overlaid on the original MRI. The second column displays the corresponding model predictions. The third column visualizes overlap between manual and predicted segmentations, with true positives (green), false positives (red), and false negatives (blue). The fourth and fifth columns show SoftMax probability maps for the thoracic volume and spine, respectively. Each voxel is assigned to the class with the highest SoftMax probability.

**Table 1 healthcare-13-02327-t001:** The computed metrics for thoracic volume segmentation in the validation dataset.

Participant	Volume [L]	DSC	HD [mm]	HD95 [mm]	Precision	Recall
1	10.10	0.89	54.38	27.19	0.83	0.96
2	6.03	0.94	27.19	1.77	0.95	1.00
3	2.42	0.87	179.41	25.50	0.78	1.00
4	2.68	0.91	25.72	3.33	0.93	1.00
5	3.02	0.94	25.50	2.36	0.94	1.00
6	4.93	0.94	25.50	1.67	0.96	1.00
7	2.42	0.90	25.55	25.50	0.89	1.00
Mean [range]	4.51 [2.42–10.10]	0.91 [0.87–0.94]	51.89 [25.50–179.41]	12.47 [1.67–27.19]	0.90 [0.78–0.96]	0.99 [0.96–1.00]

**Table 2 healthcare-13-02327-t002:** The computed metrics for 3D T1-T12 spinal length in the validation dataset.

Participant	Length [cm]	DSC	HD [mm]	HD95 [mm]	Precision	Recall
1	30.6	0.79	55.43	27.56	0.68	0.96
2	28.4	0.85	27.25	4.51	0.74	1.00
3	22.8	0.83	26.04	4.71	0.71	0.99
4	17.6	0.87	8.33	3.33	0.77	1.00
5	19.6	0.84	13.44	4.71	0.72	1.00
6	21.5	0.86	25.77	25.50	0.76	1.00
7	20.2	0.88	25.61	3.33	0.78	1.00
Mean [range]	23.0 [17.6–30.6]	0.85 [0.79–0.88]	25.98 [8.33–55.43]	10.523 [3.33–27.56]	0.74 [0.68–0.78]	0.99 [0.96–1.00]

**Table 3 healthcare-13-02327-t003:** Comparison of manual and model results for thoracic volume.

Participant	Manual Volume [L]	Model Volume [L]	% Difference
1	8.79	10.10	+14.9
2	5.77	6.03	+4.5
3	1.87	2.42	+29.4
4	2.49	2.68	+7.1
5	2.83	3.02	+6.7
6	4.76	4.93	+3.6
7	2.16	2.42	+12.0
Mean [range]	4.10 [1.87–8.79]	4.51 [2.42–10.10]	11.0 [+3.6–+29.4]
Cohen’s *d* = 1.01; 95% CI: [0.08, 1.95]			

**Table 4 healthcare-13-02327-t004:** Comparison of manual and model results for 3D spinal length.

Participant	Manual Volume [L]	Model Volume [L]	% Difference
1	24.6	30.6	+24.4
2	25.0	28.4	+13.6
3	19.1	22.8	+19.4
4	17.4	17.6	+1.1
5	20.0	19.6	−2.0
6	20.6	21.5	+4.4
7	16.7	20.2	+20.1
Mean [range]	20.5 [16.7–25.0]	23.0 [17.6–30.6]	11.6 [−2.0–+24.4]
Cohen’s *d* = 1.07; 95% CI: [0.12, 2.03]			

## Data Availability

The raw data supporting the conclusions of this article will be made available by the authors on request. The primary research question of this longitudinal study has not been analyzed yet. Data cannot be shared out of the EU for legal reasons.

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
