# Peer review of "Quantification of Thoracic Volume and Spinal Length of Pediatric Scoliosis Patients on Chest MRI Using a 3D U-Net Segmentation"

_healthcare, 2025, doi:10.3390/healthcare13182327_

Round 1
Reviewer 1 Report
Comments and Suggestions for Authors
The authors investigated the accuracy of an automatic segmentation model using a 3-D U-Net convolutional neural network on MRI images of the thoracic region from 19 children at risk of developing AIS and 19 healthy young adults to create, based on ground truth data of thoracic volume and spinal length. Thoracic volume and spinal length were calculated with an error of approximately 11–12%. The study suggested the potential for clinical application in evaluating the progression of thoracic deformity in AIS patients without radiation exposure. This study provides valuable data for clinicians and researchers in this field. However, several revisions may be necessary.
(1) Could you describe what kinds of specialty and how many years of experience five operators have in their jobs?
(2) Could you how the strength of the magnetic field of the MRI in this study?
(3) I agree with the authors' point that there was no medical exposure when MRI is used. On the other hand, MRI has limitations in that it requires movement restriction and is costly. For what types of cases and when stage do you expect it to be used?
Author Response
Reviewer #1:
Comments and Suggestions for Authors
The authors investigated the accuracy of an automatic segmentation model using a 3-D U-Net convolutional neural network on MRI images of the thoracic region from 19 children at risk of developing AIS and 19 healthy young adults to create, based on ground truth data of thoracic volume and spinal length. Thoracic volume and spinal length were calculated with an error of approximately 11–12%. The study suggested the potential for clinical application in evaluating the progression of thoracic deformity in AIS patients without radiation exposure. This study provides valuable data for clinicians and researchers in this field. However, several revisions may be necessary.
(1) Could you describe what kinds of specialty and how many years of experience five operators have in their jobs?
Author’s response: 5 Master students in Medical Imaging, trained by an orthopaedic researcher (MD) with extensive experience in spine and chest imaging.
(2) Could you how the strength of the magnetic field of the MRI in this study?
Author’s response: 1.5 Tesla.
Changes to text: p.3, line 92 ‘Philips 1.5 Tesla Achieve dStream’.
(3) I agree with the authors' point that there was no medical exposure when MRI is used. On the other hand, MRI has limitations in that it requires movement restriction and is costly. For what types of cases and when stage do you expect it to be used?
Author’s response: We agree MRI is a costly and has limitations. We expect to use this to screen the thoracic volume of Adolescent Idiopathic Scoliosis (AIS). These patients often already get an MRI for medical screening for neural axis abnormalities, that are more prevalent in AIS [1,2].
References
- Benli, I.T.; Uzümcügil, O.; Aydin, E.; Ateş, B.; Gürses, L.; Hekimoğlu, B. Magnetic Resonance Imaging Abnormalities of Neural Axis in Lenke Type 1 Idiopathic Scoliosis. Spine (Phila Pa 1976) 2006, 31, 1828–1833, doi:10.1097/01.brs.0000227256.15525.9b.
- Inoue, M.; Minami, S.; Nakata, Y.; Otsuka, Y.; Takaso, M.; Kitahara, H.; Tokunaga, M.; Isobe, K.; Moriya, H. Preoperative MRI Analysis of Patients With Idiopathic Scoliosis. Spine (Phila Pa 1976) 2005, 30, 108–114, doi:10.1097/01.brs.0000149075.96242.0e.
Reviewer 2 Report
Comments and Suggestions for Authors
Dear Authors,
Thank you for your innovative work applying a 3-D U-Net to the automated segmentation of thoracic volume and spine length from pediatric scoliosis patients on chest MRI. Your study addresses a critical clinical imperative: radiation-free, quantitative measurement of thoracic deformity in AIS highly amenable to longitudinal observation and evaluation of treatment effect.
Nonetheless, several issues are in need of explanation. Overestimation of thoracic capacity and spine length (mean +11% and +12%, respectively) appears to be related to problems with the definition of the inferior boundary and the high inter-slice gaps (20–25 mm), which break 3D continuity. The application of kidney location and T12 morphology to define boundaries introduces anatomical variability that may affect segmentation accuracy, especially in child development. In addition, the COM-based method of spinal length estimation can be slice orientation and gap dependent and potentially overestimate length values in areas of kyphosis. For more robust 3-D length estimation, we suggest the utilization of anterior-posterior vertebral landmarks. Because the training set was small (n=31), the model could be tested on a greater and more representative population, including one of actual AIS patients. It would be better to define the ground truth segmentation protocol and consider the impact of slice gaps on performance in 3D models.
Best regards
Author Response
Reviewer #2
Dear Authors,
Thank you for your innovative work applying a 3-D U-Net to the automated segmentation of thoracic volume and spine length from pediatric scoliosis patients on chest MRI. Your study addresses a critical clinical imperative: radiation-free, quantitative measurement of thoracic deformity in AIS highly amenable to longitudinal observation and evaluation of treatment effect.
Nonetheless, several issues are in need of explanation. Overestimation of thoracic capacity and spine length (mean +11% and +12%, respectively) appears to be related to problems with the definition of the inferior boundary and the high inter-slice gaps (20–25 mm), which break 3D continuity.
The application of kidney location and T12 morphology to define boundaries introduces anatomical variability that may affect segmentation accuracy, especially in child development.
In addition, the COM-based method of spinal length estimation can be slice orientation and gap dependent and potentially overestimate length values in areas of kyphosis.
For more robust 3-D length estimation, we suggest the utilization of anterior-posterior vertebral landmarks. Because the training set was small (n=31), the model could be tested on a greater and more representative population, including one of actual AIS patients. It would be better to define the ground truth segmentation protocol and consider the impact of slice gaps on performance in 3D models.
Author’s response: Thank you for your review. We agree that the high inter-slice gaps and inferior body definition probably contributed to the overestimation of the thoracic capacity and spine length. In the discussion, we also mention this limitation (p.8, line 227 – 234; p. 8, line 271 – 286). The dataset had the inherent limitation of this wide slice gap, and we would also suggest that in future studies this gap is decreased. We also suggest that future studies should use a more solid definition of this lower border. Your critique on our COM-method is also correct, and we also suggest the utilization of anterior-posterior vertebral landmarks (p.8, line 288 – 299). In future studies, we want to test the protocol on a bigger population, including actual AIS patients. Moreover, the ground truth segmentations will be improved, with a more narrow slice gap.
Changes to text: None, we believe these limitations are already extensively discussed in the discussion section, with suggestions for future studies.
Reviewer 3 Report
Comments and Suggestions for Authors
Dear Authors,
The topic of your research study entitled “Quantification of Thoracic Volume and Spinal Length of Pediatric Scoliosis Patients on Chest MRI using a 3-D U-Net Segmentation” brings into attention the possibility to use the 3-D U-Net CNN to monitor the progression of chest deformation in patients with scoliosis whereas no radiation is applied.
- I suggest you provide in the Introduction section more information about the gap in the literature and then link it to the aim of your research along with its novelty or the difference in the approach in comparison with others from the literature.
- Besides the Institutional Review Board Statement I suggest to also provide the same information in Material and Methods section.
- Please state clearly the type of study you conducted both in the abstract and the Material and Methods section.
- Regarding the inclusion and exclusion criteria, you have to state them clearly in the manuscript.
- In the Results section I recommend you give more information regarding your results and also explain why what you expected was the same/different as the obtained results.
- In your study you gave averages of the differences: Chest volume: 11% average difference between automatic vs. manual (GT); Spinal length: 12% average difference, but no indication of the variability (e.g., standard deviation, confidence intervals) or whether these differences were statistically significant. Therefore, including an effect size measure such as Cohen’s d would help quantify the magnitude of differences between automatic and manual segmentations, complementing accuracy metrics and clarifying the clinical relevance of the findings. This would complement metrics like DSC and HD, offering clearer insight into the clinical relevance of the deviations.
- In the Discussion section, I suggest you compare more your results with other from the current literature.
- I also suggest you adding the strengths and limitations of the study you conducted and adding them in a concise, clear manner.
- I recommend also adding 3-5 take-home messages and emphasizing the precision and availability of using this approach.
Good luck!
Author Response
Reviewer #3:
Dear Authors,
The topic of your research study entitled “Quantification of Thoracic Volume and Spinal Length of Pediatric Scoliosis Patients on Chest MRI using a 3-D U-Net Segmentation” brings into attention the possibility to use the 3-D U-Net CNN to monitor the progression of chest deformation in patients with scoliosis whereas no radiation is applied.
- I suggest you provide in the Introduction section more information about the gap in the literature and then link it to the aim of your research along with its novelty or the difference in the approach in comparison with others from the literature.
Author’s response: Thank you for this suggestion.
Changes to text: We added two sentences mentioning this knowledge gap and linking it to the aim: p.2 line 60 – 61 and p.2 line 74 – 76.
- Besides the Institutional Review Board Statement I suggest to also provide the same information in Material and Methods section.
Author’s response: We agree.
Changes to text: Added to p.2-3, line 89 – 91.
- Please state clearly the type of study you conducted both in the abstract and the Material and Methods section.
Author’s response: Thank you for bringing this to our attention. We performed a proof of concept study.
Changes to text: p.1, line 22 ‘In this proof-of-concept study’, p.3, line 108 – 109 ‘This study was intended as a proof-of-concept.
- Regarding the inclusion and exclusion criteria, you have to state them clearly in the manuscript.
Author’s response: We added these to the revised manuscript.
Changes to text: p.2, line 85 – 88. ‘Subjects could be included if they had no medical history of spinal or neuromuscular disease. The young sisters were part of the baseline of the prospective EARLYBIRD study, more detailed in- and exclusion criteria can be found in this study [22].’
- In the Results section I recommend you give more information regarding your results and also explain why what you expected was the same/different as the obtained results.
Author’s response: We believe this should be done in the discussion section, were we extensively discussed and compare all results.
Changes to text: None.
- In your study you gave averages of the differences: Chest volume: 11% average difference between automatic vs. manual (GT); Spinal length: 12% average difference, but no indication of the variability (e.g., standard deviation, confidence intervals) or whether these differences were statistically significant. Therefore, including an effect size measure such as Cohen’s d would help quantify the magnitude of differences between automatic and manual segmentations, complementing accuracy metrics and clarifying the clinical relevance of the findings. This would complement metrics like DSC and HD, offering clearer insight into the clinical relevance of the deviations.
Author’s response: We performed a proof-of-concept study, and mainly want to see if this technique works. We do not expect it to work perfectly. However, we added Cohen’s and the 95% CI now to table 3 and 4, and in-text p.5 lines 178 – 181. These show a high Cohen’s D value, and thus overestimation and room for improvement of the model.
Changes to text: p.5 lines 178 - 181 and table 3 and 4
- In the Discussion section, I suggest you compare more your results with other from the current literature.
Author’s response: We agree such a comparison is necessary, but this is already done in our discussion (p. 7, line 213 – 225).
Changes to text: None.
- I also suggest you adding the strengths and limitations of the study you conducted and adding them in a concise, clear manner.
Author’s response: Strengths and limitations are extensively discussed in the discussion.
Changes to text: none.
- I recommend also adding 3-5 take-home messages and emphasizing the precision and availability of using this approach.
Author’s response: We are not sure is this is MDPI format.
Changes to text: We added 3 take home messages, p. 9 – 10, lines 326 – 331.
Reviewer 4 Report
Comments and Suggestions for Authors
The authors have carried out a study that tackles an important clinical issue (radiation-free monitoring of thoracic and spinal deformities in scoliosis) and applies a modern deep learning approach (3D U-Net). However, the following comments are useful for authors considerations:
- The small dataset in this study that was carried out with only 19 at-risk girls and 19 adults. It is critical that the authors review the limitations of the generalizability due to the small sample, age mismatch between groups (children vs. adults), and lack of external validation. Add a section in the discussion stating this limitation as well as providing an implication for clinical practice and researchers. Also, due to this limitation, please do not overstate the conclusion and town down the strength of your findings both in the abstract and in the main text.
- Manual segmentation is described as the ground truth (GT). Who performed these? Were multiple raters involved? If yes, what was the inter-rater variability? If not, this introduces bias, since CNN is being benchmarked against one rater only.
- Also, While DSC, HD, precision, and recall are standard, the large HD for chest and for spine is concerning, suggesting segmentation outliers. The authors should comment on cases where segmentation failed.
- Please adhere to reporting EQUATOR guidelines relevant to this observational study design. Provide a filled-in checklist for EQUATOR checklist.
- Is it possible to cross-validate or externally validate for additional strengthening of this study findings.
Author Response
Reviewer #4:
The authors have carried out a study that tackles an important clinical issue (radiation-free monitoring of thoracic and spinal deformities in scoliosis) and applies a modern deep learning approach (3D U-Net). However, the following comments are useful for authors considerations:
- The small dataset in this study that was carried out with only 19 at-risk girls and 19 adults. It is critical that the authors review the limitations of the generalizability due to the small sample, age mismatch between groups (children vs. adults), and lack of external validation. Add a section in the discussion stating this limitation as well as providing an implication for clinical practice and researchers. Also, due to this limitation, please do not overstate the conclusion and town down the strength of your findings both in the abstract and in the main text.
Author’s response: The limitation was added and conclusion was rewritten.
Changes to text: p.9, line 303 – 304 and line 320 – 321.
- Manual segmentation is described as the ground truth (GT). Who performed these? Were multiple raters involved? If yes, what was the inter-rater variability? If not, this introduces bias, since CNN is being benchmarked against one rater only.
Author’s response: 5 master students in medical imaging, trained by an orthopedic resident. Inter-rater variability was not assessed, since we were focused on the proof-of-concept of the 3-D U-Net CNN.
- Also, While DSC, HD, precision, and recall are standard, the large HD for chest and for spine is concerning, suggesting segmentation outliers. The authors should comment on cases where segmentation failed.
Author’s response: We agree, future studies should have a more narrow slice interval and better boundary definitions to prevent outliers.
Changes to text: None, we already discussed the cases in in lines 257 – 269.
- Please adhere to reporting EQUATOR guidelines relevant to this observational study design. Provide a filled-in checklist for EQUATOR checklist.
Author’s response: Thank you for pointing this out. We now filled in the STARD-Checklist.
Changes to text: STARD-checklist was added to the submission.
- Is it possible to cross-validate or externally validate for additional strengthening of this study findings.
Author’s response: Yes, it is, and we would like to do that in future studies.
Round 2
Reviewer 3 Report
Comments and Suggestions for Authors
Dear authors,
Thank you for incorporating reviewers` recommendations in your manuscript and for your thorough revision. These changes enhanced the scientific rigor and clarity of your work and made your manuscript suitable for publication.
Reviewer 4 Report
Comments and Suggestions for Authors
Thanks for the revision.